# Laboratory testing and on-site storage are successful at mitigating the risk of release of foot-and-mouth disease virus via production of bull semen in the USA

**Anne Meyer**[1]*, **Jay Weiker**[2], **Rory Meyer**[2]

**1** Episystemic, Marcy l'Etoile, France, **2** National Association of Animal Breeders/Certified Semen Services, Inc., Madison, Wisconsin, United States of America

* anne.meyer@episystemic.fr

**Data Availability Statement:** All relevant data are within the paper and its Supporting information files.

**Funding:** The author(s) received no specific funding for this work.

## Abstract

Thousands of frozen bovine semen doses are produced daily in the US for domestic use. An incursion of foot-and-mouth disease (FMD) in the country would pose strong challenges to the movements of animals and animal products between premises. Secure supply plans require an estimation of the risk associated with target commodities and the effectiveness of mitigation measures. This study presents the results of a quantitative assessment of the risk of release of FMD virus from five of the largest commercial bull studs in the US via contaminated frozen processed semen. The methodology from a previous study was adapted to better fit the US production system and includes more recent data. Two models were combined, a deterministic compartmental model of FMD transmission and a stochastic risk assessment model. The compartmental model simulated an FMD outbreak within a collection facility, following the introduction of a latent-infected bull. The risk of release was defined as the annual likelihood of releasing at least one frozen semen batch, defined as the total amount of semen collected from a single bull on a given collection day, containing viable FMD virus. A scenario tree was built using nine steps leading from the collection to the release of a contaminated batch from a given facility. The first step, the annual probability of an FMD outbreak in a given facility, was modeled using an empirical distribution fitted to incidence data predicted by five models published between 2012 and 2022. An extra step was added to the previously published risk pathway, to account for routine serological or virological surveillance within facilities. The results showed that the mitigation measures included in the assessment were effective at reducing the risk of release. The median annual risk of release from the five facilities was estimated at less than 2 in 10 billion ($1.5 \times 10^{-10}$) in the scenario including a 30-day storage, routine genome detection assays performed every two weeks and RT-PCR testing of the semen. In this scenario, there was a 95% chance that the risk of release would be lower than 0.00041. This work provides strong support to the industry for improving their response plans to an incursion of FMD virus in the US.

**Competing interests:** The authors have declared that no competing interests exist.

# 1 Introduction

With a gross production value estimated at above 100 billion USD for 2021 [1], the livestock industry is a key sector for the US economy, both for food security and for income generation. An incursion of foot-and-mouth (FMD) virus in the country, which is currently free from the disease, would have devastating consequences in many areas such as food production, job security and animal welfare. Previous modelling studies estimated the losses that would be generated by an FMD outbreak in the US in the order of one to two hundred billion dollars [2–7], depending on the location of the index case and the extent of the outbreak. In addition to the culling of animals, FMD outbreaks would be managed using movement stand-stills for susceptible animals and animal products [8]. However, thousands of bovine semen breeding doses are produced every day in the US and shipped domestically to sustain the dairy production industry. An interruption of the supply would lead to major difficulties in the dairy sector, where most inseminations are carried out artificially. Obtaining movement permits during an outbreak for commodities which are considered safe is therefore a key challenge for the industry. Secure supply plans have been designed for this purpose in the past ten years by different livestock sectors. The pork sector was particularly active in these developments with its Secure Pork Supply Plan, given the growing threat to the industry posed by the African Swine Fever pandemic [9]. Documented risk assessments are critical to support these plans and assess their validity, by presenting the risk associated with the movements of animals and animal products from disease-free premises during an outbreak, such as weaned pigs [10] and ready-to-eat pork [11]. In a similar fashion, a secure supply plan for the distribution of frozen bovine semen will be key to ensure business continuity in the dairy sector. The present manuscript presents a quantitative assessment of the risk of release of FMD virus from some of the largest bull studs in the US via contaminated frozen bovine semen.

# 2 Material and methods

## 2.1 Approach

The methodology was adapted from the methodology described by Meyer et al. [12]. Briefly, this quantitative risk assessment was based on combining two models, a deterministic compartmental model of FMD transmission and a stochastic risk assessment model. While the 2017 paper considered one bull stud populated with FMD-vaccinated bulls in Israel, the present study considers five large US collection facilities, where vaccination is not practiced. Therefore, all bulls were considered fully susceptible to FMD in the present assessment. The rest of the section focusses on the differences from the previously published model, and a description of the components which are identical is not repeated in this manuscript.

## 2.2 Compartmental model of FMD transmission

FMD transmission between animals was modeled at the barn level to simulate the progression of an outbreak within a closed herd. The compartmental, deterministic model of FMD transmission published in Meyer et al. [12] was developed further to better suit the US context. This model simulated the temporal dynamics of the spread of FMD virus within a typical barn housing 65 bulls in individual pens, including mount animals (i.e., bovines used to collect semen from the donor bulls). While mount animals are typically steers, intact bulls or female cattle are also used. Mount animals are submitted to the same biosecurity rules and disease testing protocols as intact bulls; therefore they are not treated differently in the model.

The effective daily contact rate between individual animals is a key parameter of such a model. Published data exist for group housing, for example, a study in a 1,000-cow dairy herd

by Carpenter et al. [13], but data on transmission in individual housing is limited. An experimental study with calves housed individually in contiguous pens showed that transmission from the inoculated calves to their neighbors did not occur [14]. According to internal data from the facilities, a bull is collected on average two days per week. Each bull is in daily contact with his 2 neighbors, and, on collection days, he is also in contact with the other bulls collected that day, in the collection arena (around 20 animals). In this context and considering all these direct contacts as effective in terms of FMD virus transmission, we estimated an average of 8 effective daily direct contacts per bull.

A study in unvaccinated bulls suggested that FMD virus is excreted in semen even before it is detectable in other fluids [15–17]. Therefore, the latent stage was divided into two sub-stages, a fully latent stage, and a second latent stage where the virus is present in semen, but the animal still does not contribute to within-barn transmission. The transmission coefficients for transitions from one stage to the next stage were the mean values of the pan-serotype disease phases in cattle compiled by Yadav et al. [18]. They are presented in Table 1.

The compartmental model was built in R using the *deSolve* package [19]. The simulation was prompted by the introduction of the virus in a fully susceptible population (i.e., one animal is exposed and is in the first latent sub-stage) and run for 60 days.

The following outputs were extracted for use in the risk assessment model described below:

- The duration $t_{max}$ of the outbreak (i.e., number of days since the introduction of the virus until all bulls become immune)

- The daily prevalence of bulls infectious via semen (second latent stage, subclinical stage and clinical stage) during the outbreak

- The daily prevalence of bulls with detectable serum antibodies during the outbreak. Published data indicate that antibodies are detectable in serum from 5 to 7 days post-infection, depending on the isotype [20–22]. Therefore, we considered that 75% of clinical bulls and all recovered bulls had detectable antibodies.

- The daily prevalence of bulls with detectable viremia during the outbreak. We considered that all bulls in the infectious stage had detectable viremia.

## 2.3 Risk assessment model

**2.3.1 Model structure.**   A semen batch was defined as the total amount of semen collected from a single bull on a given collection day. Usually, this would be made up of two ejaculates (i.e., two collections) from that bull. Each semen batch is processed after collection and divided into an average of 500 individual straws, called "semen doses" or "semen units". The risk assessment model estimates the risk of release *R*, i.e., the annual likelihood of releasing at least one semen batch containing viable FMD virus.

**Table 1. Transmission coefficients for the compartmental model.** Adapted from [18].

| Disease stage | Sub-stage | Mean duration (days) |
|---|---|---|
| Latent | Latent | 1 |
| | Latent but virus present in semen | 1 |
| Infectious | Subclinical | 2 |
| | Clinical | 9 |
| Recovered | | No loss of immunity |

The model was based on the risk pathway presented in Fig 1, which shows the steps leading from the collection to the release of a contaminated batch from a given facility. The assessment did not cover subsequent steps, such as the distribution and usage of the individual straws within each batch. As the probability of each step is conditional on the previous step having been realized, the probability $P$ that a given batch released from a collection facility is infected was calculated by multiplying the probability of the individual steps along each branch of the scenario tree which may lead to the release of an infected batch [12]. Also, given that the branches are independent, the individual branch probabilities were added to obtain $P$. $R$ was estimated based on a multilevel binomial process by the following expression:

$$R = 1 - (1 - P)^N$$

where $N$ is the total number of batches released annually by the five facilities under consideration. For this assessment, we considered that an average 4-barn facility produces a mean of 59 semen batches per collection day, 250 days a year, in accordance with current industry practices.

A probability distribution of $R$ under 10 scenarios (Table 2) was created by repeated random sampling from the input probability distributions (50,000 iterations). The scenarios were based on alternative versions of the 4th, 8th and 9th steps in the risk pathways, which are described in the remainder of this section.

**2.3.2 Is the semen batch collected during an FMD outbreak in the facility?.** The risk of introduction of FMD virus into a high-biosecurity collection facility is expected to be very low, even in the context of FMD virus spreading in the US. Indeed, the probability of introduction of the virus through infected animals (cattle or wildlife), contaminated vehicles, or workers is very low. Airborne transmission also remains possible, depending on the distance to neighboring barns and other farms. The annual probability of an FMD outbreak in a given facility ($H$), was modeled using an empirical distribution fitted to the data on predicted incidence of FMD outbreaks from five models [23–27]. More details about the estimates used are available in S1 Appendix. We derived $P_1$, the probability that an individual semen batch is collected during an FMD outbreak, from $H$ via the following expression:

$$P_1 = \frac{H * t_{max}}{365}$$

**2.3.3 Does at least one bull show clinical signs before the end of storage?.** The likelihood of asymptomatic infections in unvaccinated adult cattle is low. Experimental studies have shown that 100% of unvaccinated animals display clinical signs after infection [28–30]. Older studies are not referenced here but demonstrated similar results. Reports from FMD outbreaks in the UK showed that 3 out of 3 infected cattle herds in the 2001 outbreaks, and 9 out of the 11 in the 2007 outbreaks, showed some clinical signs, while the two herds without clinical signs were at-risk epidemiological contacts examined and culled at an early stage of the infection [31, 32]. The worst-case scenario would occur when a semen batch is collected at the very beginning of an outbreak from a latently infected bull. Even in such a situation, most infected bulls would have entered the clinical stage of the disease by the end of the semen storage period on site, in both storage scenarios (see more details about storage below). Given these data, the probability that all bulls in the collection facility display an asymptomatic infection until the end of semen storage was considered negligible. To account for this highly unlikely event, the probability that at least one bull in the facility shows clinical signs before

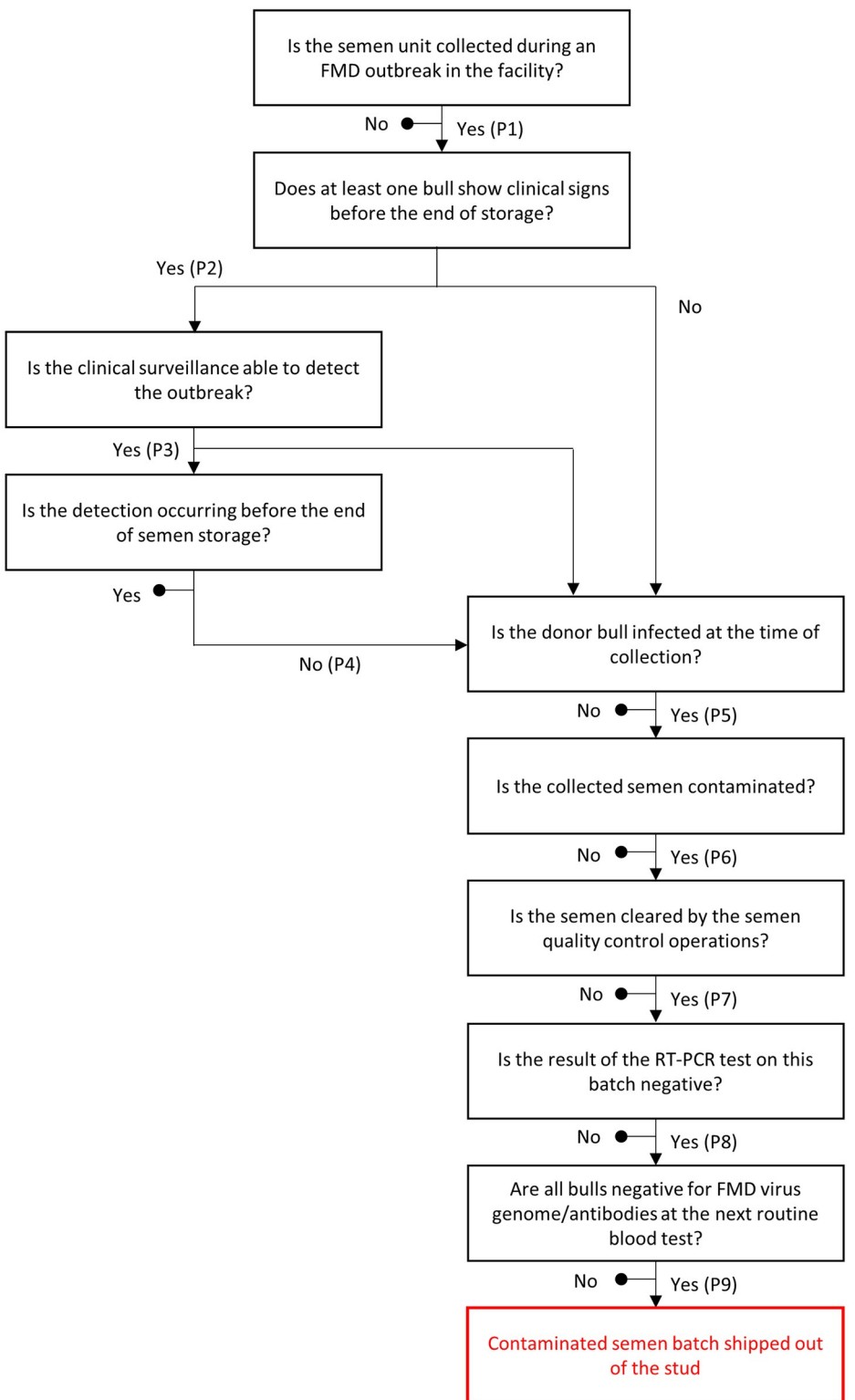

**Fig 1. Risk pathway for the release of contaminated bull semen batches.** This pathway was adapted from the one published by Meyer et al. [12].

**Table 2. Scenarios modelled in this FMD quantitative risk assessment.**

| Serum testing | Semen testing | Storage duration | Scenario identifier |
|---|---|---|---|
| Antibody detection assays, every two weeks | No testing | 14 days | 1 |
| | | 30 days | 2 |
| Genome detection assays, every two weeks | 100% of semen batches | 14 days | 3 |
| | | 30 days | 4 |
| | No testing | 14 days | 5 |
| | | 30 days | 6 |
| No | 100% of semen batches | 14 days | 7 |
| | | 30 days | 8 |
| | No testing | 14 days | 9 |
| | | 30 days | 10 |

the end of semen storage ($P_2$) was modeled as a Beta distribution with mean 0.95 and standard deviation 0.03.

**2.3.4 Is clinical surveillance able to detect the outbreak?.** Clinical surveillance, based on the detection of abnormal signs by routine observation of the animals, is performed by the facility staff. As clinical signs of FMD in unvaccinated cattle are relatively specific, involving pyrexia, ptyalism, lameness, vesicles in the mouth and/or the interdigital spaces, the sensitivity of the clinical surveillance for FMD is expected to be very good. A study of the performance of clinical monitoring of herds during the 2001 UK outbreaks estimated the sensitivity of this type of surveillance at 97.6% (with a confidence interval between 96.7 and 98.3%) [33]. The probability $P_3$ that an outbreak in a collection facility is detected, given that at least one bull shows clinical signs, was modeled using a Beta distribution with mean 0.975 and standard deviation 0.04.

**2.3.5 Is the detection occurring before the end of semen storage?.** To account for the current uncertainty in the procedures which would be followed by bull studs should FMD be present in the US, two scenarios were envisaged for this step. In these scenarios, the duration of semen storage, between the time of collection and the shipment to domestic cattle farms, was set at 14 and 30 days (variable *storage*), respectively. The first value reflects the information included in Secure Supply plans and the testing strategies already in place in collection centers for other diseases. The second value is aligned with the recommendations for export of bull semen from non-free areas by the Terrestrial Animal Health Code [34].

Given that bulls are housed individually and under intense scrutiny by facility staff, it was assumed that the detection of an FMD outbreak would occur on average two days after the onset of clinical signs [35, 36]. We added this delay to the expected incubation period of 6 days (sum of the latent and subclinical stages) to obtain an average duration between the introduction of the virus and the detection of the outbreak (or expected time to detection, $\lambda$) of 6 + 2 = 8 days. The outbreak detection was assumed to follow a Poisson process with parameter $\lambda$, meaning that the probability of the event "outbreak detection" occurring per unit of time was assumed to be constant [37]. The Poisson process also assumes that the probability of the event is independent of the number of those events having occurred before, but in this context, we were only interested in the first detection event. Under a Poisson process, the Gamma distribution with shape 1 and scale $\lambda$ estimates the time until the first detection event has occurred [37]. $P_4$ is the probability that the outbreak has *not* been detected by the time the semen is released. To incorporate uncertainty around the expected time to detection, $P_4$ was modelled as a Beta-PERT distribution with minimum 0, maximum 0.5 and mode $\bar{P}_4$. The mode was

calculated according to the duration of the storage period using the Gamma distribution:

$$\bar{P}_4 = 1 - P(Gamma(1, \lambda) \leq \ storage)$$

**2.3.6 Is the donor bull infected at the time of collection?.** The daily within-herd prevalence of bulls infectious via semen was extracted from the compartmental model, as described above. These data were fitted to an empirical distribution to model the probability that the donor bull is infectious on the day of collection, given that an undetected outbreak of FMD is occurring in the facility ($P_5$).

**2.3.7 Is the collected semen contaminated?.** In the absence of vaccination, the probability of excretion of virus in semen is high, given the generalized nature of the infection. Previous studies found FMD virus in the semen of 3, 6 and 15 bulls out of 4, 6 and 20 infected, unvaccinated bulls, respectively [15–17]. Therefore, the probability that the semen batch collected contains FMD virus, given that the donor bull is infected ($P_6$), was modeled using a Beta distribution with mean 0.80 and standard deviation 0.087.

**2.3.8 Is the semen cleared by the semen quality control operations?.** After routine observation, semen quality control operations are the second stage of health surveillance in collection facilities. Depending on facilities, semen ejaculates are evaluated by a laboratory technician, by computer-assisted semen analysis (CASA), or both. The probability $P_7$ that the semen is cleared for release by quality control operation, given that the donor bull is infected, depends on two parameters, $M$ and $S_m$. $M$ is the probability that the semen microscopic characteristics are abnormal, given that the donor bull is infected. A study showed that three out of four experimentally infected bulls presented abnormal semen characteristics, such as low sperm count, low sperm viability, or high abnormality rate [15]. Data for these bulls were only available for the clinical phase of the disease. There were no data on semen abnormalities during the pre-clinical phase of the disease, when semen may already contain FMD virus. We assumed that semen microscopic abnormalities would be less common during that phase. Therefore, M was modeled by a Beta distribution with mean 0.50 and standard deviation 0.14.

Parameter $S_m$ is the probability that abnormal microscopic characteristics are detected by the semen quality control operations, or sensitivity of these operations. Very few data were available in the literature to estimate this parameter. A study showed that the sensitivity of CASA was 88%, using the diagnostic made by experienced technicians as the reference [38]. Another study showed that correlation between laboratory technicians varied from 38% to 92%, depending on the criteria (mobility, viability, concentration), but no further details are available regarding assessment performance [39]. These studies support the hypothesis that semen quality control operations are generally thorough. Therefore, $S_m$ was modeled by a Uniform distribution between 0.75 and 1. We also assumed that the specificity (i.e., the probability of not detecting an abnormality in the normal semen) was 100%, so that, $P_7$ was estimated as follows:

$$P_7 = 1 - M * S_m$$

**2.3.9 Is the result of the RT-PCR test on this semen batch negative?.** Contaminated semen may be released if the reverse transcription-polymerase chain reaction (RT-PCR) test performed on it produces a false negative result. We assumed that a RT-PCR assay could be performed on a proportion $p_{ej}$ of the batches. This procedure is not currently routinely performed in the US. Therefore, we included two scenarios in our assessment, where no testing is performed ($p_{ej} = 0$) and 100% of batches would be tested, respectively ($p_{ej} = 1$). The sensitivity

of RT-PCR assays used for the detection of FMD virus genome in blood and serum is very high [40, 41]. In the absence of published evidence on the capacity of these assays to detect FMD virus genome in semen, the sensitivity of the RT-PCR performed on semen ($S_p$) was modeled using a Beta distribution with mean 0.95 and standard deviation 0.04. The probability that the result of the semen RT-PCR test is negative, given that the batch contains FMD virus ($P_8$) is given by:

$$P_8 = 1 - p_{ej} * S_p$$

**2.3.10 Are all bulls negative for FMD virus genome/antibodies at the next routine blood test?.** This step was added to the risk assessment model to better fit the US context, and was not present in the model published by Meyer et al. [12]. We considered that blood would be collected from bulls at collection facilities and tested for the presence of antibodies against FMD virus or viral genome on a routine basis, to improve the sensitivity of outbreak surveillance and align with regulatory surveillance requirements. The assays were assumed to be conducted on serum, as the sensitivity of RT-PCR in particular was reported to be higher on serum than whole blood samples [42].

The routine accounted for in the model was an assay performed on serum samples from 83% of bulls in the facility every two weeks. Hence, one or two rounds of sampling and testing would be performed during the storage period of each semen batch, depending on the storage duration (14 or 30 days). The calculation used in this work assumed random sampling of 83% of the bulls from the facility. Any sampling method targeting higher-risk bulls, for example those having more frequent contacts, would result in a reduction in risk compared with the estimate presented here. Two alternative scenarios were considered, one where sera are tested with an antibody detection test, and one where sera are tested with a genome detection test.

*Antibody detection on serum.* The proportion of bulls with detectable antibodies at the time of serum collection was extracted from the compartmental model. These data were fitted to an empirical distribution to model the probability $T_{a1}$ that a given bull has antibodies, given that an undetected outbreak of FMD is occurring in the facility. The distribution of $T_{a2}$ was adjusted for the second antibody test performed in the scenarios with a 30-day storage, to account for the increase in antibody prevalence at that time. The sensitivity of ELISA assays to detect antibodies against non-structural proteins of FMD virus in cattle serum is generally high, but there are significant variations between test kits [43, 44]. The sensitivity also increases with time after infection, but a constant value was used here for simplicity. The sensitivity of the antibody assay ($S_a$) was modeled using a Uniform distribution between 0.92 and 1.

*Viral genome detection on serum.* Similarly, we considered the effect of routine testing of bulls for viral genome. This practice was envisaged given the high probability of false positive test results expected under the antibody testing scenario, due the imperfect specificity of the ELISA assays. The proportion of bulls with a detectable viremia at the time of serum collection was also extracted from the compartmental model. These data were fitted to an empirical distribution to model the probability $T_{g1}$ that a given bull is viremic, given that an undetected outbreak of FMD is occurring in the facility. The distribution of $T_{g2}$ was also adjusted for the second test performed in the scenarios with a 30-day storage. The sensitivity of real-time reverse-transcription polymerase chain reaction (rRT-PCR) assays to detect FMD virus genome in cattle serum is very high, up to 100% [45], even in field conditions [46]. Their very high specificity also makes these assays particularly suited for routine testing of a large number

of animals. The sensitivity of the assay for genome detection ($S_g$) was modeled as a Beta-PERT distribution with minimum 0.95, maximum 1.00 and mode 0.98.

The probability $P_9$ that all bulls tested are negative at the next routine serum test was given by the following equation, where j = 1 in the 14-day storage scenarios, j = 2 in the 30-day storage scenarios, k = $g$ for genome detection tests and k = $a$ for antibody detection tests:

$$P_9 = \prod_{i=1}^{j} (1 - T_{ki} * S_k)^{p_{blood}*pop}$$

Given that this procedure is not currently routinely performed in the US, we also calculated the risk of release under a scenario where routine blood sampling is not performed (i.e., considering $P_9 = 1$).

## 2.4 Sensitivity analysis

A sensitivity analysis was performed using linear regression to assess the relative importance of the different input parameters. Both the input parameters used as predictors in the model and the risk of release used as the output variable were scaled. The model coefficients provide the change in risk associated with one standard deviation increment from the mean of the input parameters.

## 3 Results

The size of each compartment is shown in Fig 2. The duration $t_{max}$ of the outbreak was estimated at 51 days, after which all bulls were in the recovered compartment. The estimated distribution of the annual probability of release of at least one contaminated semen batch from the five facilities (R) is presented in Table 3 and Fig 3. Summary values and graphical representations of the parameters used in the model are presented in S1 Appendix. The distribution of the risk of release for scenarios 1 to 6 presented a long upper tail, with very few values up to 0.99, despite a very small median value. Antibody testing of serum was most effective at reducing the risk, and with routine antibody detection performed every two weeks (scenarios 1 and 2), the median risk of release was extremely low (under $1.0 \times 10^{-11}$).

Results showed that testing for viral genome was expected to provide less confidence in disease freedom. With a 30-day storage and routine genome detection assays performed every two weeks (scenario 6), the median risk of release was $5.0 \times 10^{-9}$, with a 95th percentile at 0.011. Under these conditions, adding RT-PCR testing of the semen batches helped lower the risk of release: the median risk of release was $1.5 \times 10^{-10}$, with a 95th percentile at 0.00041 under scenario 4. Finally, the median risk of release was always higher than 0.0057 in scenarios without routine serum testing (scenarios 7 to 10), even with RT-PCR testing of the semen batches.

Sensitivity analysis was conducted on the risk of release associated with scenario 4 (RT-PCR testing of semen, 30 days of semen storage, routine genome detection testing of serum). The results for increasing input parameters by one standard deviation from the mean (Fig 4) show that the parameters which influence most the probability of a contaminated semen batch were (by decreasing order of influence): the probability that a given bull has detectable viremia at the time of serum sampling ($T_{g1}$ and $T_{g2}$), the prevalence of infectious bulls at the time of semen collection ($P_5$), the sensitivity of the RT-PCR performed on semen ($S_p$), the risk of introduction of FMD virus in the herds (H), and the probability that the outbreak has *not* been detected by the time of release ($P_4$). While the plot shows the results specifically for Scenario 4, similar results were obtained for other scenarios.

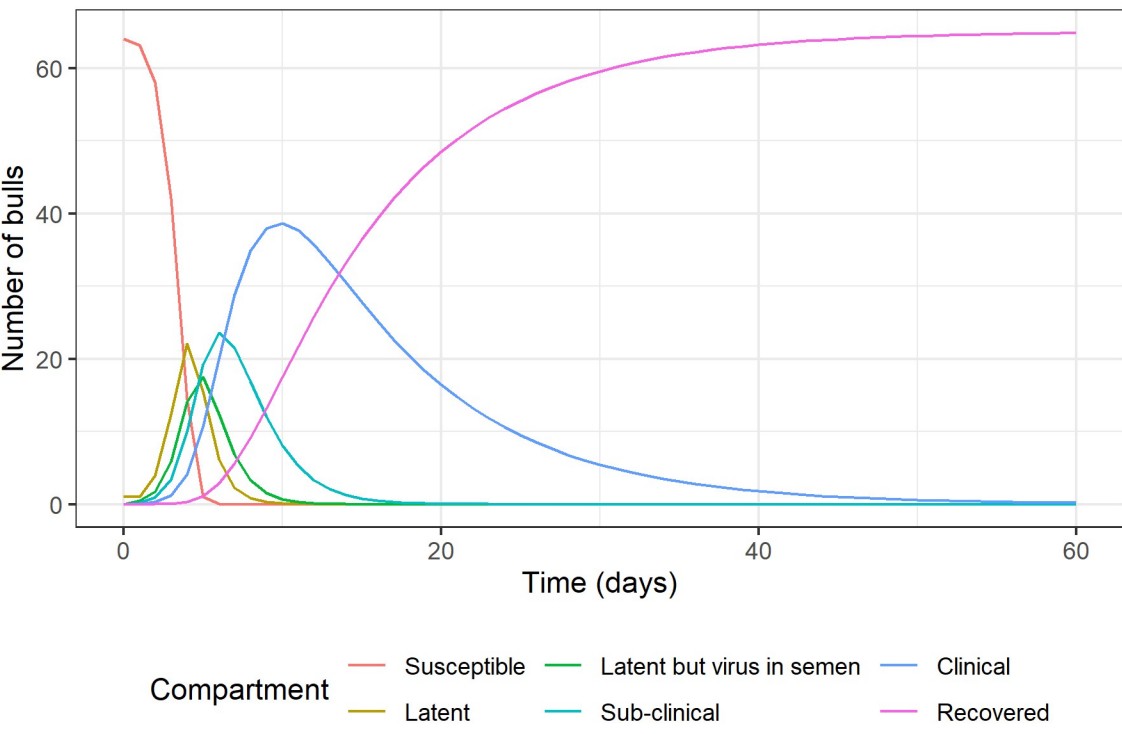

**Fig 2. Graphical output of the FMD compartmental model.**

## 4 Discussion

The results show that while the three mitigation measures included in this assessment are effective at reducing the risk of release (routine serum testing, testing of the semen batches and a longer storage duration), their application in combination appears to be required for a low risk of release. Laboratory testing of sera, via either antibody or genome detection tests, appeared to be a critical mitigation step, as the median risk of release was higher than 0.005 in all scenarios where it was not implemented (scenarios 7 to 10).

**Table 3. Estimated risk of release and years to release of FMD virus in bull semen from the five facilities under consideration (R).**

| Scenario identifier | Annual probability of release of at least one contaminated batch | | Years until one contaminated batch is released | |
|---|---|---|---|---|
| | Median | 95th percentile | Median | 95th percentile |
| 1 | $< 1.0 \times 10^{-11}$ | $3.9 \times 10^{-3}$ | Over 100 billion | 257 |
| 2 | $< 1.0 \times 10^{-11}$ | $< 1.0 \times 10^{-11}$ | Over 100 billion | Over 100 billion |
| 3 | $1.2 \times 10^{-6}$ | $9.9 \times 10^{-3}$ | 860,000 | 101 |
| 4 | $1.5 \times 10^{-10}$ | $4.1 \times 10^{-4}$ | 7 billion | 2,439 |
| 5 | $4.1 \times 10^{-5}$ | 0.22 | 24,452 | 5 |
| 6 | $5.0 \times 10^{-9}$ | $1.1 \times 10^{-2}$ | 198 million | 89 |
| 7 | $8.9 \times 10^{-3}$ | 0.14 | 113 | 7 |
| 8 | $5.7 \times 10^{-3}$ | $9.5 \times 10^{-2}$ | 176 | 11 |
| 9 | 0.25 | 0.94 | 4 | 1 |
| 10 | 0.17 | 0.86 | 6 | 1 |

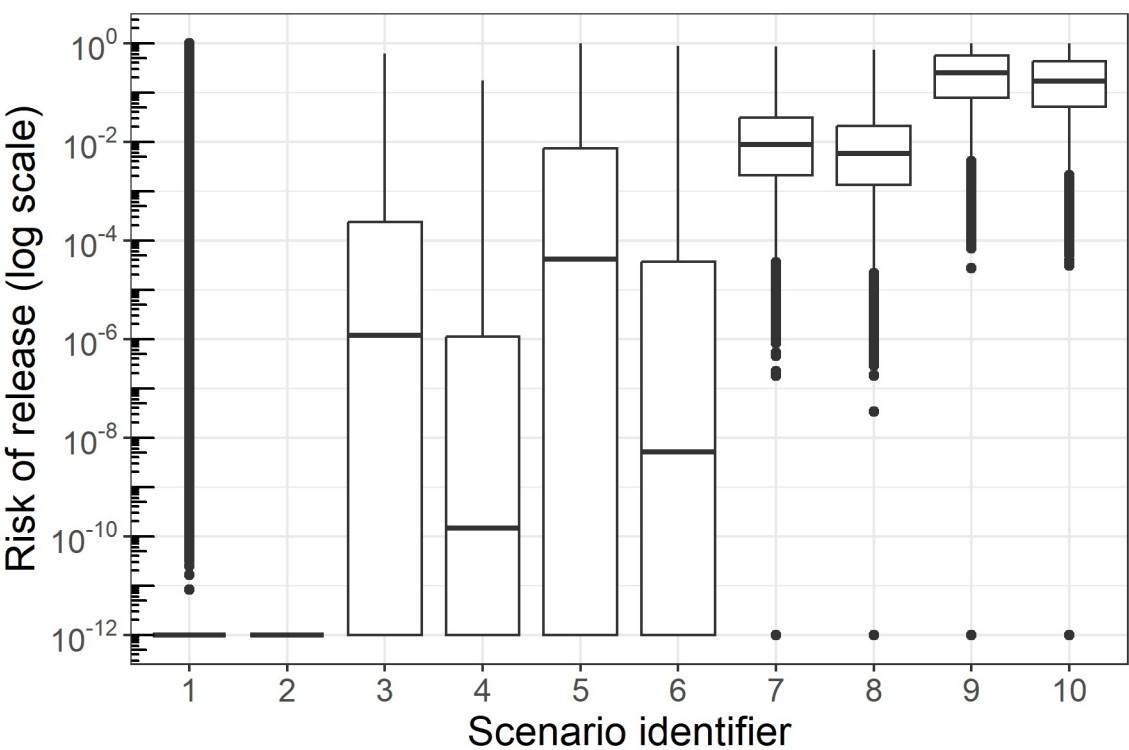

**Fig 3. Estimated risk of release of FMD virus in bull semen from the five facilities under consideration (R).**

Antibody testing of serum was most effective at reducing the risk, given the persistence of antibodies in serum long after the infectious phase. However, the regular false positive results expected from large numbers of ELISA assays may be difficult to manage in practice. With a specificity of 97%, as estimated in previous work [43, 44], three false positive results are expected for 100 samples tested. Therefore, genome detection testing may be more suitable in practice, given the very high specificity of the test. Genome detection testing provided less confidence in disease freedom, due to the limited duration of the viremic phase, but can be considered effective when coupled with the other mitigation steps. Under the most likely scenario including a 30-day storage, routine genome detection assays performed every two weeks and RT-PCR testing of the semen batches (scenario 4), the median annual risk of release from the five facilities was estimated at less than 2 in 10 billion ($1.5 \times 10^{-10}$). In this scenario, there was a 95% chance that the risk of release would be lower than 0.00041.

Sensitivity analysis showed that some parameters were more influential than others. In particular, the three most important parameters in scenario 4 were extracted from the compartmental model ($T_{ag-1}$, $T_{ag-2}$ and $P_5$). These parameters depend themselves on the input parameters for that model (e.g., effective contact rate, duration of each disease stage, seed size). The stage transition parameters used in this study differ from the serotype-O parameters previously estimated by Mardones et al. [47], where the mean latent stage duration was 3.6 days and the mean infectious stage duration 4.4 days. While those values were previously used in the US to simulate the spread of FMD using the North American Animal Disease Spread Model (NAADSM), current practices have changed to use the more recent estimates instead. In our model, we used values summarized by Yadav et al. [18], with a shorter latent period (mean 2 days) and a longer infectious period (mean 11 days). We were specifically interested

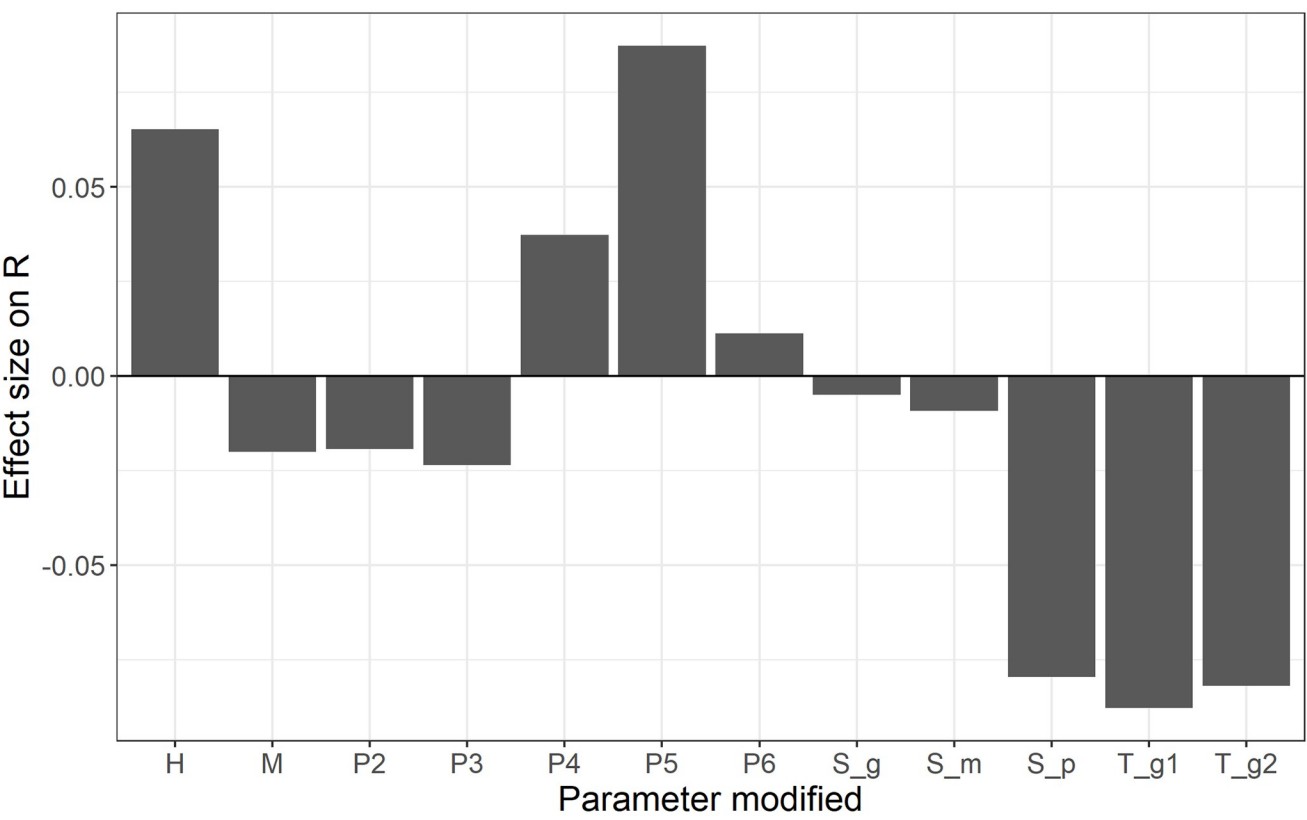

**Fig 4. Sensitivity analysis on the risk of release under Scenario 4, estimated using a linear regression model with scaled predictors.**

in the presence of FMD virus in semen, for which blood may be used as a proxy. Transmission experiments have shown that a shorter latent period and a longer infectious period are typically obtained when calculated from proxy measures of infection, such as virus detection from blood, nasal fluid or oesophageal-pharyngeal fluid [48]. However, using a direct measure of infectiousness by assessing the transmission of the disease to another animal, these authors found a longer latent period and a shorter infectious period, in alignment with figures reported by Mardones et al. [47].

While the duration of each disease stage is well documented for FMD, it is more difficult to estimate within-herd infection dynamics in the specific context of a semen collection facility. Authors previously estimated effective contact rates for FMD using field data [49], by expert elicitation [50], using simulation models [51] or based on observational experience [13]. The estimate used in this study (8 effective contacts per day) did not account for indirect contacts between animals via airborne exposure, fomites and caretakers for example. It also did not account for potential direct contact between bulls led by staff in the alleys on the way from and to the collection arena for example. Data on within-farm dynamics of outbreaks occurring in unvaccinated animals are scarce, and very little data were available to support an effective contact rate estimate including these transmission pathways in an individual housing setup, even when considering experimental studies. Transmission between individually housed calves did not occur in a study by Bouma et al. [14], while most in-contact calves became infected in a group-housing context [52]. This suggests that airborne transmission is not an effective transmission route in calves. However, figures obtained from calves may not apply to adult cattle,

given that transmission between young animals appears less effective than between adult cattle, as suggested by comparing excretion titers and secondary infections [53, 54]. Other studies in calves showed that a significant proportion of the transmission in a group-housing context occurred via the environment, contaminated by the secretions and excretions of infectious animals [55, 56]. Fomite-based transmission is expected to occur in collection facilities as well but could not be quantified here. Consequently, the estimate used in the analysis did not include indirect pathways and may be a biased estimated of the true contact rate.

Interestingly, the probability of clinical signs ($P_2$) and the probability that clinical surveillance detects the outbreak ($P_3$) were not among the most influential parameters. This can be explained by the fact that the variance used for these parameters was relatively small, and that sensitivity analysis was based on increasing mean values by a unit of standard deviation. We estimated that the probability that at least one bull in the facility would show clinical signs was very high, based on experimental studies and field reports. Sub-clinical infection does occur at the individual level, and creates a significant risk of introducing disease in importing countries and herds [57]. However, we did not find any reported evidence of unvaccinated adult cattle herds remaining asymptomatic for the entire course of the outbreak. Studies showed that this may be possible in calves, given that they sometimes remain clinically normal or develop only very subtle signs over the course of the infection [53, 54, 58]. Notably, evidence often comes from experimental studies rather than field reports, and the caveat of using the former is that infectious doses are high compared to those in naturally occurring transmission. Regarding clinical detection, there was limited data to estimate the parameter distribution. Given that bulls are high-value animals, housed individually, and with a staff-to-animal ratio larger than in a standard cattle farm, we used a relatively high probability of clinical detection in the presence of signs. However, an independent assessment of the performance of clinical surveillance in these settings would be useful to support this assumption.

In terms of seed size, the compartmental model assumed that FMD would be introduced in the facility via one bull in the latent stage. This scenario corresponds to the physical introduction of a new bull, or fomite exposure of a resident bull via the attending veterinarian for example. It does not cover situations where the virus would be introduced via feed or a large fomite contamination. However, a recent modelling study suggested that the seed size was not a very influential parameter in within-herd transmission models where density-dependent transmission is assumed [59].

Our compartmental model was not set up to account for variability and uncertainty in input parameter values nor in state transitions and only provides a deterministic assessment. In particular, the model does not account for the fact that transmission of and infection with FMD virus are stochastic processes, which may fail in the real world. All bulls in the facility became infected in our model, while field studies show that a proportion of animals usually remains uninfected during outbreaks. The deterministic model was deemed sufficient for the purpose of this assessment, but further work could address this limitation by including parameter distributions instead of point estimates, as well as introducing stochasticity in the state transition processes. The risk assessment model on the other hand, adapted from a previous iteration, accounted for variability and uncertainty in the input parameters. Many of these parameters were updated from the original version to better fit the US context and to include more recent knowledge.

Another key parameter was the probability of introduction of FMD virus into collection facilities. Disease spread between farms in previously FMD-free areas has been previously assessed using real-world data, such as the 2001 outbreaks in the UK [60, 61] and in Argentina [62], respectively. However, there is little published data on the incidence of FMD outbreaks in high-biosecurity facilities such as bull or boar studs. Previous reports have shown that

biosecurity breaches may also occur in such facilities, for example the 1997 outbreaks of classical swine fever in Denmark [63], but it is difficult to assess the risk difference quantitatively. In addition, the US having been long free of FMD, there were no available observational data on incidence of FMD outbreaks in the country. Given the specificities of the US livestock production system compared to other countries, incidence data from other countries appear not very applicable. Therefore, we elected to gather incidence data estimated by mathematical models which have been specifically fitted to the US livestock production system. Limitations of this approach remain, given that in three models, the outputs encompassed cattle as well as other species (sheep, swine), and none of the estimates specifically reflected the high biosecurity level of semen collection facilities. It is also notable that the number of infected premises predicted by these models often presented a bimodal distribution, with most outbreaks affecting a very small number of farms, and a few larger outbreaks. This translated into relatively high mean estimates of the farm-level incidence, higher than the median estimates from the model simulations. In this work, we chose to use the mean estimates to better account for the occurrence of these rare but larger outbreaks.

Finally, the multinomial process used to estimate $R$ is assuming independence between semen batches. This is unlikely to hold in practice for batches produced by the same facility. It is expected that both the probability of detecting the outbreak and the probability of collecting contaminated semen would increase as the outbreak progresses in the facility. In the absence of documented evidence on FMD outbreaks in semen collection facilities, it is difficult to assess the effect of interdependency on $P$ and $R$. This is a structural limitation of using scenario trees for risk assessments. The approach used in this work is inspired from the methodology promoted by the World Organization for Animal Health to conduct import risk analysis for animals and animal products [64]. It is primarily calibrated at estimating the probability of occurrence of the first disease event but does not account well for subsequent disease spread and interdependence between disease events. Indeed, it is not possible to account for infection dynamics in such scenario tree models directly, and additional methods are required, such as coupling them with disease transmission models. Recommendations to improve how risk assessments are conducted in the animal health field have been proposed, but they primarily concern qualitative assessments [65, 66]. Regardless, our study provides supporting evidence for the cattle artificial insemination sector in the US to consolidate their disease preparedness plans, in case FMD virus should be introduced in the country.

## Supporting information

**S1 Appendix. Additional model inputs and outputs used to estimate the risk of release of foot-and-mouth disease virus via production of bull semen in the USA.**
(PDF)

## Acknowledgments

The authors thank Dr James Meronek for the information on semen production infrastructure and processes provided during the design phase of this work.

## Author Contributions

**Conceptualization:** Anne Meyer, Jay Weiker, Rory Meyer.

**Formal analysis:** Anne Meyer.

**Investigation:** Rory Meyer.

**Methodology:** Anne Meyer, Rory Meyer.

**Project administration:** Jay Weiker, Rory Meyer.

**Resources:** Jay Weiker, Rory Meyer.

**Supervision:** Jay Weiker, Rory Meyer.

**Validation:** Anne Meyer, Jay Weiker, Rory Meyer.

**Visualization:** Anne Meyer.

**Writing – original draft:** Anne Meyer.

**Writing – review & editing:** Anne Meyer, Jay Weiker, Rory Meyer.

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
