## [Decision Letter · Decision Letter 0]

17 Aug 2023

PONE-D-23-24651Quantitative assessment of the risk of release of foot-and-mouth disease virus via production of bull semen in the USAPLOS ONE

Dear Dr. Meyer,

Thank you for submitting your manuscript to PLOS ONE. After careful consideration, we feel that it has merit but does not fully meet PLOS ONE’s publication criteria as it currently stands. Therefore, we invite you to submit a revised version of the manuscript that addresses the points raised during the review process.

We look forward to receiving your revised manuscript.

Kind regards,

Junyuan Yang

Academic Editor

PLOS ONE

Journal Requirements:

Reviewers' comments:

Reviewer's Responses to Questions

**Comments to the Author**

1. Is the manuscript technically sound, and do the data support the conclusions?

Reviewer #1: Yes

Reviewer #2: Yes

2. Has the statistical analysis been performed appropriately and rigorously? 

Reviewer #1: Yes

Reviewer #2: Yes

3. Have the authors made all data underlying the findings in their manuscript fully available?

Reviewer #1: No

Reviewer #2: Yes

4. Is the manuscript presented in an intelligible fashion and written in standard English?

Reviewer #1: Yes

Reviewer #2: Yes

5. Review Comments to the Author

Reviewer #1: Quantitative assessment of the risk of release of foot-and-mouth disease virus via production of bull semen in the USA

Anne Meyer, Jay Weiker, Rory Meyer

This is an interesting modelling study on use of laboratory tests to mitigate the risk of release of FMDV via a bull stud. I would suggest to include the main mitigation methods in the title. The title should tell the story (tell them what you going to tell them). It is quite difficult to understand the methodology the text could be more precise. For the laboratory tests the authors should use laboratory jargon (antigen and genome testing are different) and veterinary jargon (lockdown is a human medicine control measure not a veterinary). The compartmental model is a deterministic model and therefore 100% of the animals will become infected. In a stochastic model every individual has a probability of becoming infected and sometimes infection fails. A stochastic model resembles much better what is happening in the field where antibodies against non-structural proteins are often only present in a proportion of the animals whereas they were all present during the outbreak. Still the deterministic model is probably sufficient for the purpose, but it would be good to mention the limitations.

I think the estimate of the sensitivity of clinical inspection is too high, I would use a slightly lower estimate and a wider confidence interval. It would be interesting to see if it would end up in the top 5 of variables that have the highest influence on the outcome.

There are more papers on quantifying transmission, especially performed for the use in models, it would be good to use them (I think I listed all that involved transmission in non-vaccinated cattle).

Comments:

Abstract:

Generatio spotanea is not happening. So, infection should start somewhere, this is not mentioned in the abstract. The abstract is vague, be precise, be concise.

Introduction:

Line 39: Lockdown was used during the corona epidemic but does not cover "movement stand-still" which is normally used in veterinary medicine. A lockdown during coronal included the shut-down of opera, music festivals etcetera, but cows do not attend opera. People were allowed to move during a lockdown, even fly to foreign countries. During a "movement stand-still" livestock is kept on the farm.

Line 51 - 53: I would find the risk of causing a new outbreak more interesting than the risk of releasing FMD virus.

Materials and methods:

Line 67: The US context, is not specific. Barnsize, number of barns per facility etc. is more specific, e.g. a table with the relevant information would be nice.

Line 69: What is a teaser on a bull stud? In other farming systems these are castrated male animals, but I assume there are no castrated bulls on a bull stud.

Line 72: For individual housing see: Bouma, A., Dekker, A., de Jong, M. C. M. 2004. No foot-and-mouth disease virus transmission between individually housed calves. Vet. Microbiol. 98(1); 29-36.

Line 76: You mean 8 daily 'direct' contacts. The number of indirect contacts must be much higher, especially if you would include airborne transmission.

Line 77 - 78: You should also include: Cottral, G. E., Gailiunas, P., Cox, B. F. 1968. Foot-and-mouth disease virus in semen of bulls and its transmission by artificial insemination. Arch. Gesamte Virusforsch. 23(4); 362-377. And Sharma, G. K., Subramaniam, S., De, A., Das, B., Dash, B. B., Sanyal, A., Misra, A. K., Pattnaik, B. 2012. Detection of foot-and-mouth disease virus in semen of infected cattle bulls. Indian J. Anim. Sci. 82(12); 1472-1476. There is not a lot of data on excretion of FMDV in semen but ignoring >50% of the published papers is strange. Later you mention both papers.

Line 89: Tmax? It is a bit strange to assume that all bull become immune. In epidemiology it is assumed that an infection enters a state in which the effective reproduction number Re = 1 = R0 x (1 - recovered fraction). So, the recovered fraction will be 1 - 1/R0. For R0 = 4, then the recovered fraction will be 0.75. Estimates of R0 in endemic settings are 1.2 - 1.6, but in a bull stud it might be higher. In within pen transmission studies R0 was over 10, but many between pen transmission studies show low R0 values.

Line 107 - 108: What do you mean with antigen assay. Do you mean a complement fixation test or ELISA to detect viral antigen in the sample? (antigen is the protein that is immunogenic and can be detected by ELISA or complement fixation test; virus is capable of infecting cells and can be detected by virus isolation; genome can be tested by RT-PCR). In figure 1 you mention PCR test (which is incorrect as it an RNA virus so you would perform an RT-PCR test), so you do not do antigen detection, but genome detection.

Line 110: A batch contains 200 - 800 straws (estimate from an interview in a newspaper, I do not know if this is accurate, but the author should know and mention this), I assume that even if 1 straw is removed you consider the batch released. Or are the collection facility and the distribution facility 2 separate entities?

Line 124: It is a valid assumption that an AI centre would have a high biosecurity status, but in the 1997 CSF outbreak in the Netherlands an AI centre became infected and transmitted CSF to many farms. The moment they knew CSF was present they quickly removed some old boars for slaughter, but due to the haste biosecurity was breached……. The personal that were responsible for the boars helped them on the truck that moved them to the slaughterhouse, but after entering the truck they entered the AI facility through the same door as was used to remove the boars. So, they did not use the showering facility to enter the place as they did every morning……… Is the probability really low?

S1: Trying to understand I did the calculation 143/28779 = 0.004968901 (neither for the other models). This does not match the numbers in the table. Are the numbers correct? I'm missing the number of replicates in each model, as I understand that you used the outcomes of different runs for the empirical distribution of H.

Line 133: What is the value of tmax? When I use P1 x 365 / H = 0.00034 x 365 / 0.0024 = 51.7 days (the unit is nowhere defined, I assume it is days). I see this is mentioned in the results.

Line 135: In most experimental studies 10 000 bovine ID50 is injected into the tongue, then 100% clinical disease is seen. But in studies with other routes of infection, clinical disease might be absent (see e.g. Sutmoller, P., Olascoaga, R. C. 2002. Unapparent foot and mouth disease infection (sub-clinical infections and carriers): implications for control. Rev. Sci. Tech. 21(3); 519-529). What is the sensitivity of the model for this assumption of 100% clinical disease (compare e.g. with 80%).

Line 152: The sensitivity of clinical detection might be biased. The persons responsible for clinical detection did the estimation. The estimate for the 2006 UK outbreak was much lower. Cattle in the field were observed over the fence, clinical disease was missed. It would be good to do a sensitivity analysis on this parameter (e.g. 50% sensitive)

Line 164: "intense scrutiny" is this true? When the bull is not used for collecting semen it is only fed, and not inspected on a daily basis.

Line 176: I searched Wikipedia for Beta PERT distribution but could not find it. Do you mean a normal PERT distribution?

Line 215 - 225 and elsewhere: Replace PCR by RT-PCR

Line 227: Be specific. "Are all bulls negative for FMD genome at the next routine blood test?"

Line 237: What is meant by "The use of either an antigen or an antibody test was considered"? Be precise.

Line 248: You refer to Brocchi et al. who validated ELISA tests to detect antibodies against non-structural proteins. That is a good choice, but you should mention that in line 241 "Antibody ELISA to detect antibodies against non-structural proteins". In non-vaccinated cattle the sensitivity of a good NS ELISA is high but increases with the time after infection. In the Brocchi paper all 5 sera collected 7-14 dpi in non-vaccinated cattle were positive.

Line 249: replace "Antigen assay on serum" by "Genome detection in serum".

Line 256: sensitivity of the RT-PCR is high when compared to virus isolation. But the probability of scoring a positive sample in a bull that is infected is important. This is also high in non-vaccinated cattle. But referring to sensitivity is not that relevant if virus isolation is the gold standard.

Line 289: How many false positive results are expected? 3 every 100 samples? Be precise.

Line 290 and elsewhere: Replace "antigen" by "genome" or "RT-PCR"

Line 298 - 306: Do I understand correctly that clinical detection is less important than RT-PCR, this is probably due to the limited SD that is given to clinical detection. In the sensitivity analysis only 1 SD difference is tested.

Line 313: Replace "Serological surveillance" by "Laboratory testing of sera" (serological surveillance refers to evaluation of antibodies and is not used for RT-PCR testing).

Line 350 - 353: Perhaps the author should evaluate the following papers it gives information on relation between transmission rate in calves that are separated (Bouma et al. 2004) and calves that mix freely (Orsel et al. 2005):

Bouma, A., Dekker, A., de Jong, M. C. M. 2004. No foot-and-mouth disease virus transmission between individually housed calves. Vet. Microbiol. 98(1); 29-36.

Orsel, K., Dekker, A., Bouma, A., Stegeman, J. A., de Jong, M. C. M. 2005. Vaccination against foot and mouth disease reduces virus transmission in groups of calves. Vaccine 23(41); 4887-4894.

For the risk of transmission by the environment by indirect contact (not airborne as Bouma et al. showed that calves in the same room did not transmit FMDV).

Bravo de Rueda, C., de Jong, M. C., Eble, P. L., Dekker, A. 2015. Quantification of transmission of foot-and-mouth disease virus caused by an environment contaminated with secretions and excretions from infected calves. Vet. Res. 46; 43.

Colenutt, C., Brown, E., Nelson, N., Paton, D. J., Eblé, P., Dekker, A., Gonzales, J. L., Gubbins, S. 2020. Quantifying the Transmission of Foot-and-Mouth Disease Virus in Cattle via a Contaminated Environment. mBio 11(4); e00381-20.

Quantification of the transmission rate in the incubation period was studied by:

Graves, J. H., McVicar, J. W., Sutmoller, P., Trautman, R. 1971. Contact Transmission of Foot-and-Mouth Disease from Infected to Susceptible Cattle. J. Infect. Dis. 123(4); 386-391.Orsel, K., Bouma, A., Dekker, A., Stegeman, J. A., de Jong, M. C. M. 2009. Foot and mouth disease virus transmission during the incubation period of the disease in piglets, lambs, calves, and dairy cows. Prev. Vet. Med. 88(2); 158-163.

Charleston, B., Bankowski, B. M., Gubbins, S., Chase-Topping, M. E., Schley, D., Howey, R., Barnett, P. V., Gibson, D., Juleff, N. D., Woolhouse, M. E. 2011. Relationship between clinical signs and transmission of an infectious disease and the implications for control. Science 332(6030); 726-9.

Reviewer #2: Quantitative assessment of the risk of release of foot-and-mouth disease virus via production of bull semen in the USA

Aim: The present manuscript presents a quantitative assessment of the risk of release of FMD virus from some of the largest bull studs in the US via contaminated frozen bovine semen.

https://doi.org/10.3390/ani11061697 : this paper will help you in introduction and discussion

6. PLOS authors have the option to publish the peer review history of their article (what does this mean?). If published, this will include your full peer review and any attached files.

Reviewer #1: No

Reviewer #2: No

---

## [Author Response · Author response to Decision Letter 0]

12 Sep 2023

All line numbers are reported as per the unmarked version of your revised paper without tracked changes.

Reviewer #1

1. This is an interesting modelling study on use of laboratory tests to mitigate the risk of release of FMDV via a bull stud. I would suggest to include the main mitigation methods in the title. The title should tell the story (tell them what you going to tell them). 

Thank you for the thorough review. We believe the manuscript has been improved and provide responses to each comment below. The title of the manuscript was modified to “Laboratory testing and on-site storage are successful at mitigating the risk of release of foot-and-mouth disease virus via production of bull semen in the USA”.

2. It is quite difficult to understand the methodology the text could be more precise. For the laboratory tests the authors should use laboratory jargon (antigen and genome testing are different) and veterinary jargon (lockdown is a human medicine control measure not a veterinary). 

This has been improved, according to comments 7 and 15 below.

3. The compartmental model is a deterministic model and therefore 100% of the animals will become infected. In a stochastic model every individual has a probability of becoming infected and sometimes infection fails. A stochastic model resembles much better what is happening in the field where antibodies against non-structural proteins are often only present in a proportion of the animals whereas they were all present during the outbreak. Still the deterministic model is probably sufficient for the purpose, but it would be good to mention the limitations.

Model limitations were expanded upon in the Discussion (lines 406-413). Also see our response to comment 14 below.

4. I think the estimate of the sensitivity of clinical inspection is too high, I would use a slightly lower estimate and a wider confidence interval. It would be interesting to see if it would end up in the top 5 of variables that have the highest influence on the outcome.

See our response under comment 21 below.

5. There are more papers on quantifying transmission, especially performed for the use in models, it would be good to use them (I think I listed all that involved transmission in non-vaccinated cattle).

See our response under comment 34 where the papers are listed.

6. Abstract: Generatio spotanea is not happening. So, infection should start somewhere, this is not mentioned in the abstract. The abstract is vague, be precise, be concise.

A sentence about the outbreak simulation following the introduction of a latent-infected bull was added. The abstract was clarified further, within the allocated word limit.

7. Line 39: Lockdown was used during the corona epidemic but does not cover "movement stand-still" which is normally used in veterinary medicine. A lockdown during coronal included the shut-down of opera, music festivals etcetera, but cows do not attend opera. People were allowed to move during a lockdown, even fly to foreign countries. During a "movement stand-still" livestock is kept on the farm.

“Lockdown” replaced with “stand-still”.

8. Line 51 - 53: I would find the risk of causing a new outbreak more interesting than the risk of releasing FMD virus.

We agree that the risk of causing a new outbreak by inseminating cattle with infectious semen would also be a useful indicator for decision making. However, this requires defining yet another set of parameters (time until insemination, infectious dose, factors affecting recipient susceptibility, etc.), which go beyond the scope of the present work.

9. Line 67: The US context, is not specific. Barn size, number of barns per facility etc. is more specific, e.g. a table with the relevant information would be nice.

The following sentence (lines 72-74) already provides this information: the model simulates the outbreak in a typical barn housing 65 bulls housed in individual pens. The compartmental model does not simulate transmission between barns, only within an infected barn.

10. Line 69: What is a teaser on a bull stud? In other farming systems these are castrated male animals, but I assume there are no castrated bulls on a bull stud.

After discussion within the team, we have changed “teaser” to “mount animals” (bovines used to collect semen from the donor), as this better reflect the current practices. Mounts can be steers, but also intact bulls or female cattle, which have the same health status than intact bulls (same biosecurity rules, same testing protocols, etc.). Sentences added in this regard at lines 74-77.

11. Line 72: For individual housing see: Bouma, A., Dekker, A., de Jong, M. C. M. 2004. No foot-and-mouth disease virus transmission between individually housed calves. Vet. Microbiol. 98(1); 29-36.

Reference added at lines 80-82. Discussion edited as well at lines 370-382.

12. Line 76: You mean 8 daily 'direct' contacts. The number of indirect contacts must be much higher, especially if you would include airborne transmission.

Corrected to “daily direct contacts” (line 86).

13. Line 77 - 78: You should also include: Cottral, G. E., Gailiunas, P., Cox, B. F. 1968. Foot-and-mouth disease virus in semen of bulls and its transmission by artificial insemination. Arch. Gesamte Virusforsch. 23(4); 362-377. And Sharma, G. K., Subramaniam, S., De, A., Das, B., Dash, B. B., Sanyal, A., Misra, A. K., Pattnaik, B. 2012. Detection of foot-and-mouth disease virus in semen of infected cattle bulls. Indian J. Anim. Sci. 82(12); 1472-1476. There is not a lot of data on excretion of FMDV in semen but ignoring >50% of the published papers is strange. Later you mention both papers.

Both references were added in this paragraph (line 88).

14. Line 89: Tmax? It is a bit strange to assume that all bull become immune. In epidemiology it is assumed that an infection enters a state in which the effective reproduction number Re = 1 = R0 x (1 - recovered fraction). So, the recovered fraction will be 1 - 1/R0. For R0 = 4, then the recovered fraction will be 0.75. Estimates of R0 in endemic settings are 1.2 - 1.6, but in a bull stud it might be higher. In within pen transmission studies R0 was over 10, but many between pen transmission studies show low R0 values.

Regarding Tmax, see response to comment 19 below. We are not assuming that all bulls become immune, but this is an expected result of the deterministic SEIR model used in this work. Introducing stochastic state transitions would be a way to improve the model, but this was beyond the scope of this work, where a simple deterministic compartmental model was deemed sufficient for our purpose. We added text in the discussion to highlight this limitation (lines 406-413).

15. Line 107 - 108: What do you mean with antigen assay. Do you mean a complement fixation test or ELISA to detect viral antigen in the sample? (antigen is the protein that is immunogenic and can be detected by ELISA or complement fixation test; virus is capable of infecting cells and can be detected by virus isolation; genome can be tested by RT-PCR). In figure 1 you mention PCR test (which is incorrect as it an RNA virus so you would perform an RT-PCR test), so you do not do antigen detection, but genome detection.

Indeed, we meant detection of the genome of FMD virus via RT-PCR. The terminology was corrected in table 1, figures and text (section 2.3.10, Results, Discussion, S1 appendix).

16. Line 110: A batch contains 200 - 800 straws (estimate from an interview in a newspaper, I do not know if this is accurate, but the author should know and mention this), I assume that even if 1 straw is removed you consider the batch released. Or are the collection facility and the distribution facility 2 separate entities?

Indeed, batches are divided into several hundreds of straws, we have added an average value at line 113 for illustration. Indeed, here we consider the risk of release at the level of a batch, without looking at the distribution and destination of individual straws within batches (see addition at lines 117-119). Several straws will be removed from each batch for testing purposes, to detect different pathogens as well as assess the semen quality.

17. Line 124: It is a valid assumption that an AI centre would have a high biosecurity status, but in the 1997 CSF outbreak in the Netherlands an AI centre became infected and transmitted CSF to many farms. The moment they knew CSF was present they quickly removed some old boars for slaughter, but due to the haste biosecurity was breached……. The personal that were responsible for the boars helped them on the truck that moved them to the slaughterhouse, but after entering the truck they entered the AI facility through the same door as was used to remove the boars. So, they did not use the showering facility to enter the place as they did every morning……… Is the probability really low?

We agree that at human failure is often the root cause of biosecurity breaches, as pointed out by the reviewer. In addition, there was no supporting data to quantify the difference between biosecurity levels in collection facilities versus regular cattle farms. Thus, the estimate used for H (probability of introduction) in our analysis does not account for potentially higher biosecurity levels and is based on the risk estimated for “regular” farms. This limitation is pointed out at lines 428-430.An additional sentence in this respect was added at lines 420-423.

18. S1: Trying to understand I did the calculation 143/28779 = 0.004968901 (neither for the other models). This does not match the numbers in the table. Are the numbers correct? I'm missing the number of replicates in each model, as I understand that you used the outcomes of different runs for the empirical distribution of H.

Regarding the calculations. For the Tildesley paper, indeed there was a typo in the mean incidence, which has now been corrected, sorry for that. For the other papers, please note that small differences in the last digit of the mean incidence are due to rounding error. The mean numbers of IP are available with several decimal digits, on which the incidence calculation was based. However, in the table we presented the number of IP as an integer, leading to these small discrepancies when doing the calculations from the figures provided in S1. Mean numbers were presented with one decimal digit in the revised version of the table to provide consistency for the reader.

Regarding the replicates. Number of replicates were not used in the parametrisation process. Unfortunately, only summary figures of the number of IP estimated by the different models were available in the papers or their supporting information, sometimes only as charts. Detailed model outputs were not available, hence the decision to use the summary estimates in our analysis. We used the five point-estimates of mean incidence to fit the empirical distribution of H, giving them equal weights. 

19. Line 133: What is the value of tmax? When I use P1 x 365 / H = 0.00034 x 365 / 0.0024 = 51.7 days (the unit is nowhere defined, I assume it is days). I see this is mentioned in the results.

The duration of the outbreak, tmax, was estimated via the compartmental model, as the time until there are no more infectious bulls in the barn. This is presented in the Methods (lines 99-100). The value obtained from the model (51 days) is then reported in the Results (line 294). The unit was added at lines 99-100.

20. Line 135: In most experimental studies 10 000 bovine ID50 is injected into the tongue, then 100% clinical disease is seen. But in studies with other routes of infection, clinical disease might be absent (see e.g. Sutmoller, P., Olascoaga, R. C. 2002. Unapparent foot and mouth disease infection (sub-clinical infections and carriers): implications for control. Rev. Sci. Tech. 21(3); 519-529). What is the sensitivity of the model for this assumption of 100% clinical disease (compare e.g. with 80%).

We agree with the figures presented in this comment. However, parameter P2 is not the proportion of bulls showing clinical signs, but the probability that at least one bull in the herd displays clinical signs. Both studies and field reports show that this value is very high in adult cattle, in other words, the probability that the infection remains asymptomatic in the entire barn is very small. Field reports from the UK support this assumption. The two herds which were clinically healthy but infected were observed and culled at early stages of the process because of being at-risk contacts, so we cannot know whether clinical signs would have developed. We have decided to decrease the mean of the distribution and increase the standard deviance to account for the fact that experimental infectious doses are not representative of field conditions. The mean outputs have slightly increased accordingly, but this parameter was still not among the most influential ones. The text was updated at lines 151-154, 162 and in the discussion (lines 383-395).

21. Line 152: The sensitivity of clinical detection might be biased. The persons responsible for clinical detection did the estimation. The estimate for the 2006 UK outbreak was much lower. Cattle in the field were observed over the fence, clinical disease was missed. It would be good to do a sensitivity analysis on this parameter (e.g. 50% sensitive)

The sensitivity of routine monitoring to detect suspect cases of FMD was estimated at more than 97% in the 2001 UK outbreaks, as reported by McLaws et al (2007) and cited in section 2.3.4. We could not find corresponding estimates for the 2007 UK outbreak in the published literature. We agree that clinical detection in cattle raised outdoors would be lower, but this is not the case in our settings. Given that bulls are high-value animals housed indoor, with a higher staff-to-animal ratio than in a standard cattle farm, we expect the sensitivity of routine monitoring to be at least as high as that in the 2007 study. Therefore, we used the mean estimated by these authors for our parameter P3. In the absence of other studies to support this, we decided to widen the confidence interval to better reflect the uncertainty, as suggested by the reviewer. To do that, we had to change the distribution from Normal to Beta, so that the higher values did not exceed 1. After this change, the parameter was still not among the most influential ones. Regardless, an independent assessment of this assumption would be useful, we mentioned it in the discussion (lines 395-399).

22. Line 164: "intense scrutiny" is this true? When the bull is not used for collecting semen it is only fed, and not inspected on a daily basis.

Regardless of whether they are collected or not, bulls are fed twice per day, and they are observed for any abnormal sign or behaviour during feeding time. Would an exotic disease outbreak occur in the country, the observation time would be increased. Therefore, we believe that the probability of detection of clinical signs is adequate.

23. Line 176: I searched Wikipedia for Beta PERT distribution but could not find it. Do you mean a normal PERT distribution?

The beta-PERT distribution is the full name of what is often simply called a PERT distribution. It got its name because its distribution function is related to that of the beta distribution. See for example the books by Vose (“Fundamentals of risk analysis and risk management” and “Risk analysis: a quantitative guide”).

24. Line 215 - 225 and elsewhere: Replace PCR by RT-PCR

All mentions of “PCR” were corrected to “RT-PCR”.

25. Line 227: Be specific. "Are all bulls negative for FMD genome at the next routine blood test?"

Title 2.3.10 modified (including both options of genome and antibody testing), as well as Figure 1.

26. Line 237: What is meant by "The use of either an antigen or an antibody test was considered"? Be precise.

We moved the sentence a bit further at the end of the paragraph (lines 256-258) and reworded it to be more precise. It now better introduces the two alternative scenarios which are described just after.

27. Line 248: You refer to Brocchi et al. who validated ELISA tests to detect antibodies against non-structural proteins. That is a good choice, but you should mention that in line 241 "Antibody ELISA to detect antibodies against non-structural proteins". In non-vaccinated cattle the sensitivity of a good NS ELISA is high but increases with the time after infection. In the Brocchi paper all 5 sera collected 7-14 dpi in non-vaccinated cattle were positive.

Text modified accordingly at lines 264-267. We pointed out that for simplicity reasons, we did not include a time-varying sensitivity parameter.

28. Line 249: replace "Antigen assay on serum" by "Genome detection in serum".

Replaced (line 269).

29. Line 256: sensitivity of the RT-PCR is high when compared to virus isolation. But the probability of scoring a positive sample in a bull that is infected is important. This is also high in non-vaccinated cattle. But referring to sensitivity is not that relevant if virus isolation is the gold standard.

We removed this sentence as it was not really useful to describe the parameter.

30. Line 289: How many false positive results are expected? 3 every 100 samples? Be precise.

We removed this point from the Results section, given that it was redundant with the interpretation provided in the Discussion. We added the number of false positive results expected at lines 335-338.

31. Line 290 and elsewhere: Replace "antigen" by "genome" or "RT-PCR"

Replaced.

32. Line 298 - 306: Do I understand correctly that clinical detection is less important than RT-PCR, this is probably due to the limited SD that is given to clinical detection. In the sensitivity analysis only 1 SD difference is tested.

Indeed, the sensitivity analysis assesses the changes in risk associated with changes of 1 SD for each parameter. This caveat is now mentioned at lines 383-386. This sensitivity analysis technique cannot be used to compare mitigation measures as such. So we cannot say that clinical detection was less important than RT-PCR, given that all scenarios included clinical detection. To compare both mitigation measures, we would need to compare the risk estimates between a scenario including clinical detection and not semen testing, and a scenario including semen testing but not clinical detection. In practice, clinical detection is very cost-effective and will always be used and promoted within collection facilities, so a scenario without it was not considered.

33. Line 313: Replace "Serological surveillance" by "Laboratory testing of sera" (serological surveillance refers to evaluation of antibodies and is not used for RT-PCR testing).

Replaced (line 331).

34. Line 350 - 353: Perhaps the author should evaluate the following papers it gives information on relation between transmission rate in calves that are separated (Bouma et al. 2004) and calves that mix freely (Orsel et al. 2005). 

- Bouma, A., Dekker, A., de Jong, M. C. M. 2004. No foot-and-mouth disease virus transmission between individually housed calves. Vet. Microbiol. 98(1); 29-36.

- Orsel, K., Dekker, A., Bouma, A., Stegeman, J. A., de Jong, M. C. M. 2005. Vaccination against foot and mouth disease reduces virus transmission in groups of calves. Vaccine 23(41); 4887-4894. 

For the risk of transmission by the environment by indirect contact (not airborne as Bouma et al. showed that calves in the same room did not transmit FMDV).

- Bravo de Rueda, C., de Jong, M. C., Eble, P. L., Dekker, A. 2015. Quantification of transmission of foot-and-mouth disease virus caused by an environment contaminated with secretions and excretions from infected calves. Vet. Res. 46; 43.

- Colenutt, C., Brown, E., Nelson, N., Paton, D. J., Eblé, P., Dekker, A., Gonzales, J. L., Gubbins, S. 2020. Quantifying the Transmission of Foot-and-Mouth Disease Virus in Cattle via a Contaminated Environment. mBio 11(4); e00381-20.

Quantification of the transmission rate in the incubation period was studied by:

- Graves, J. H., McVicar, J. W., Sutmoller, P., Trautman, R. 1971. Contact Transmission of Foot-and-Mouth Disease from Infected to Susceptible Cattle. J. Infect. Dis. 123(4); 386-391.

- Orsel, K., Bouma, A., Dekker, A., Stegeman, J. A., de Jong, M. C. M. 2009. Foot and mouth disease virus transmission during the incubation period of the disease in piglets, lambs, calves, and dairy cows. Prev. Vet. Med. 88(2); 158-163.

- Charleston, B., Bankowski, B. M., Gubbins, S., Chase-Topping, M. E., Schley, D., Howey, R., Barnett, P. V., Gibson, D., Juleff, N. D., Woolhouse, M. E. 2011. Relationship between clinical signs and transmission of an infectious disease and the implications for control. Science 332(6030); 726-9.

These papers were added to the discussion to improve the paragraph on transmission routes (Bouma et al., 2004; Orsel et al., 2005, 2009; Bravo de Rueda et al., 2015), see lines 370-382. Regarding the three last papers on transmission during the incubation period, we based our data on a recent review (Yadav et al, 2019) which estimated transmission parameters, including pre-clinical transmission (subclinical infectious stage in our model) from existing literature. We believe this review adequately covered pre-existing work. 

Reviewer #2

The present manuscript presents a quantitative assessment of the risk of release of FMD virus from some of the largest bull studs in the US via contaminated frozen bovine semen. https://doi.org/10.3390/ani11061697: this paper will help you in introduction and discussion.

Thank you for your suggestion. However, these findings come from a different genus/species, under different farming conditions, and from vaccinated animals. Therefore, we believe that they are not relevant to our study.

---

## [Decision Letter · Decision Letter 1]

24 Oct 2023

Laboratory testing and on-site storage are successful at mitigating the risk of release of foot-and-mouth disease virus via production of bull semen in the USA

PONE-D-23-24651R1

Dear Dr. Meyer,

We’re pleased to inform you that your manuscript has been judged scientifically suitable for publication and will be formally accepted for publication once it meets all outstanding technical requirements.

Kind regards,

Junyuan Yang

Academic Editor

PLOS ONE

Additional Editor Comments (optional):

Reviewers' comments:

Reviewer's Responses to Questions

**Comments to the Author**

1. If the authors have adequately addressed your comments raised in a previous round of review and you feel that this manuscript is now acceptable for publication, you may indicate that here to bypass the “Comments to the Author” section, enter your conflict of interest statement in the “Confidential to Editor” section, and submit your "Accept" recommendation.

Reviewer #2: All comments have been addressed

2. Is the manuscript technically sound, and do the data support the conclusions?

Reviewer #2: Yes

3. Has the statistical analysis been performed appropriately and rigorously? 

Reviewer #2: Yes

4. Have the authors made all data underlying the findings in their manuscript fully available?

Reviewer #2: Yes

5. Is the manuscript presented in an intelligible fashion and written in standard English?

Reviewer #2: Yes

6. Review Comments to the Author

Reviewer #2: All requested comments have been corrected so the manuscript is ready to be Accept

Thousands of frozen bovine semen doses are produced daily in the US for domestic

use. An incursion of foot-and-mouth disease (FMD) in the country would pose strong

challenges to the movements of animals and animal products between premises.

Secure supply plans require an estimation of the risk associated with target

commodities and the effectiveness of mitigation measures. This study presents the

results of a quantitative assessment of the risk of release of FMD virus from five of the

largest commercial bull studs in the US via contaminated frozen processed semen.

The methodology from a previous study was adapted to better fit the US production

system and includes more recent data. Two models were combined, a deterministic

compartmental model of FMD transmission and a stochastic risk assessment model.

The compartmental model simulated an FMD outbreak within a collection facility

7. PLOS authors have the option to publish the peer review history of their article (what does this mean?). If published, this will include your full peer review and any attached files.

Reviewer #2: **Yes: **Ahmed N F Neamat-Allah

---

## [Editor Report · Acceptance letter]

30 Oct 2023

PONE-D-23-24651R1 

Laboratory testing and on-site storage are successful at mitigating the risk of release of foot-and-mouth disease virus via production of bull semen in the USA 

Dear Dr. Meyer:

I'm pleased to inform you that your manuscript has been deemed suitable for publication in PLOS ONE. Congratulations! Your manuscript is now with our production department. 

Kind regards, 

on behalf of

Dr. Junyuan Yang 

Academic Editor

PLOS ONE